# Clarifying the Dominant Role of Crystallinity and Molecular Orientation in Differently Processed Thin Films of Regioregular Poly(3-hexylthiophene)

**DOI:** 10.3390/mi15060677

**Published:** 2024-05-22

**Authors:** Kumar Vivek Gaurav, Harshita Rai, Kshitij RB Singh, Shubham Sharma, Yoshito Ando, Shyam S. Pandey

**Affiliations:** Graduate School of Life Science and Systems Engineering, Kyushu Institute of Technology, Kitakyushu 808-0196, Fukuoka, Japan; kvgaurav2010@gmail.com (K.V.G.); harshitarai30@gmail.com (H.R.); krbs09@gmail.com (K.R.S.); shubhammudgal95@gmail.com (S.S.); yando@life.kyutech.ac.jp (Y.A.)

**Keywords:** regioregular poly(3-hexylthiophene), organic field-effect transistors, thin films, crystallinity, orientation, unidirectional floating film transfer

## Abstract

Conjugated polymers (CPs) offer the potential for sustainable semiconductor devices due to their low cost and inherent molecular self-assembly. Enhanced crystallinity and molecular orientation in thin films of solution-processable CPs have significantly improved organic electronic device performance. In this work, three methods, namely spin coating, dip coating, and unidirectional floating-film transfer method (UFTM), were utilized with their parametric optimization for fabricating RR-P3HT films. These films were then utilized for their characterization via optical and microstructural analysis to elucidate dominant roles of molecular orientation and crystallinity in controlling charge transport in organic field-effect transistors (OFETs). OFETs fabricated by RR-P3HT thin films using spin coating and dip coating displayed field-effect mobility (*μ*) of 8.0 × 10^−4^ cm^2^V^−1^s^−1^ and 1.3 × 10^−3^ cm^2^V^−1^s^−1^, respectively. This two-time enhancement in *µ* for dip-coated films was attributed to its enhanced crystallinity. Interestingly, UFTM film-based OFETs demonstrated *μ* of 7.0 × 10^−2^ cm^2^V^−1^s^−1^, >100 times increment as compared to its spin-coated counterpart. This superior device performance is attributed to the synergistic influence of higher crystallinity and molecular orientation. Since the crystallinity of dip-coated and UFTM-thin films are similar, ~50 times improved *µ* of UFTM thin films, this suggests a dominant role of molecular orientation as compared to crystallinity in controlling the charge transport.

## 1. Introduction

Semiconductor technology has developed and evolved rapidly, which has enabled the fabrication of miniaturized and efficient electronic devices. Further, from the invention of transistors (in the 1950s) to today’s scenario of integrating billions of transistors on a single chip, semiconductor technology has revolutionized electronic devices [1]. Now, time demands the development of sustainable semiconductor devices owing to the paradigm shift from hard and brittle to soft and flexible. Thanks to the discovery of organic conjugated polymers having the capability of low-cost fabrication, being light-weight and flexible, and having a natural tendency toward molecular self-assembly, which offers various applications in the field of organic electronic devices [2], such as organic field-effect transistors (OFETs) [3], organic light emitting diodes [4,5], memristors [6], organic solar cells [7], etc. Moreover, OFETs can be developed as a promising frontier, utilizing organic materials instead of traditional semiconductors, owing to their potential advantages such as flexibility, low-cost manufacturing, and compatibility with unconventional substrates, paving the way for flexible electronics for developing wearable devices. However, challenges remain in achieving a comparable performance to conventional semiconductor technologies, driving ongoing research and development in this exciting field.

To address these challenges, there is a need to understand and study various components involved in the fabrication of organic electronic devices. In this rapidly advancing field, organic semiconductors are vital for the fabrication of these devices. Among the various components involved in fabricating organic electronic devices utilizing solution-processable conjugated polymers (CPs), thin film fabrication is the most important [8,9,10,11], as the device characteristics are greatly influenced by the thin film morphology of the CP layer [12,13,14]. One of the challenges faced by organic electronic devices is managing the cost and performance. The performance of these devices depends on CPs and the nature of thin films. Further, various methods of thin film fabrication can be opted for, according to the device’s requirement; as OFETs are planar devices, a homogenous thin film with aligned CP chains along the channel direction with an edge-on orientation is preferred for a better device performance. While several common methods such as bar coating [15], mechanical rubbing [16], and friction transfer [17] impart face-on orientation in thin films and lead to mechanical damage of the film, they are thus not suitable for planar devices and cannot be used for multilayer fabrication of the active organic semiconducting layer based on pure solution-based thin film fabrication approaches. There are other processing techniques for CPs like meniscus-guided coating methods, which are simple, efficient, and low-cost methods, but there are issues related to large area fabrication, high orientation, and multilayer coating [18]. The most widely used spin coating technique though results in a thin film with crystalline domains distributed in an amorphous sea of polymeric chains resulting in hampered device performance [19,20]. Therefore, methods involving increased molecular self-assembly and better molecular alignment are highly required. To circumvent these issues, thin film fabrication techniques such as dip coating [21] can be used, which can not only harness the enhanced molecular self-assembly of CPs but also solve the huge material wastage in the case of spin coating. At the same time, the unidirectional floating-film transfer method (UFTM) [22] developed and improvised by our research group provides large area, uniform, anisotropic, and edge-on oriented thin films with the least material wastage. To access the quality of thin films fabricated by various methods in depth, optical, microstructural, and electrical characterizations are very crucial.

Owing to the CPs’ quasi-one-dimensional nature, many researchers have investigated their backbone orientation and self-assembly to enhance the optoelectronic properties of the devices [23]. Alkyl-substituted CPs exhibit good solubility in common halogenated solvents, which are often used for the fabrication of thin films and their utilization of organic electronic devices [24]. There are reports for the fabrication of thin films using different thin film fabrication methods; results are scattered in terms of the CPs and method of film fabrication under investigation. Since both the factors such as the nature of the CP and the method of the film fabrication have a profound influence on the morphology, crystallinity, and orientation of the fabricated thin films, it is difficult to pinpoint the roles of the thin film crystallinity and molecular orientation on the charge transport in general and device performance in particular. Therefore, in this present work, regioregular poly(3-hexylthiophene) (RR-P3HT) was used as a solution-processable representative organic semiconductor to fabricate OFETs, where thin films of this polymer were fabricated by different methods such as UFTM, dip coating, and spin coating. Various parameters were first optimized under each type of film fabrication method followed by their optical and microstructural characterizations. Finally, OFETs were fabricated to analyze the implications of molecular orientation, uniformity, and crystallinity on the charge transport and device performance.

## 2. Materials and Methods

### 2.1. Materials

An electronic grade RR-P3HT (M_w_ = 62 kDa; PDI = 1.75) with the molecular structure shown in Figure 1A has been procured from Lisicon, Merck, Rahway, NJ, USA, and used as a representative solution processable CP for this work. All the solvents used for cleaning the substrate, including hexane, acetone, methanol, and toluene, were acquired from Fujifilm Wako, Osaka, Japan. Some other solvents used in the process, such as super-dehydrated chloroform, octadecyltrichlorosilane (OTS) and hexamethyldisilane (HMDS), were procured from Sigma Aldrich, St. Louis, MO, USA. Super-dehydrated chloroform was used to make the RR-P3HT solution of varying concentrations. Ethylene glycol (Eg) and glycerol (Gl) were purchased from Fujifilm Wako, Japan, and were used to prepare viscous liquid substrates of varying viscosities. CYTOP (CTL-809 M) and its diluting solvent (CTSOLV 180) were purchased from AGC, Chiba, Japan, and used as it is without any further purification. Chemicals such as octadecyl trichlorosilane (OTS), hexamethyldisilazane (HMDS), and CYTOP, a fluoropolymer, were used as surface modifiers on the glass and Si/SiO_2_ substrates before thin film fabrication of RR-P3HT under different film fabrication methods.

### 2.2. Thin Film and Device Fabrication

For the fabrication of thin films, three different techniques such as dip coating, spin coating, and UFTM have been used in this study: For the optimization of the fabrication conditions, thin films were fabricated on glass substrates, and finally, devices were fabricated under optimized conditions on the Si/SiO_2_ (300 nm) substrates for the investigation of charge transport by OFET fabrication. Substrates were cleaned by hexane wiping follSowed by sonication in acetone, methanol, and chloroform for 10 min each. After sonication, the substrates were subjected to different surface treatments such as UV–ozone, HMDS, OTS, and CYTOP, which were used to modify the surface of the substrates. The cleaned substrates were subjected to UV–ozone for 10 min before being used for film fabrication. For HMDS surface treatment, the 20% HMDS solution was prepared in dry hexane at room temperature followed by dipping of the substrates in this solution for 10 min at 45 °C and its cleaning by sonication in hexane for 10 min. Finally, substrates were annealed at 100 °C for 30 min to remove any residual solvents before the fabrication of thin films of RR-P3HT. OTS surface treatment was conducted by putting the substrates into 20 mM OTS solution in toluene for 36 h at room temperature to form a self-assembled monolayer (SAM). These substrates were then sonicated in toluene for 10 min and annealed at 180 °C for 30 min. In the case of using CYTOP as a surface modifier, the CYTOP solution (1:3, *v*/*v*) was prepared by its dilution using a CT-SOLV solvent, which was spin-coated on the substrates for 60 s at 2000 rpm, followed by annealing in an argon atmosphere at 180 °C for 1 h. Optical, surface, and microstructural characterizations were performed after casting the thin films of RR-P3HT using different techniques onto the modified glass substrate as discussed above followed by annealing at 120 °C. The OFET fabrication was carried out in bottom gate top contact (BGTC) device architecture on the Si/SiO_2_ substrates as shown in Figure 1B. After casting the thin films on surface modified Si/SiO_2_ substrate, annealing was performed at 120 °C. Finally, nickel shadow masks were used to pattern source/drain electrodes (50 nm) using thermal vapor deposition of gold under a 10^−6 ^ Torr.

#### 2.2.1. Spin Coating

Spin coating is the simplest and the most widely used technique to fabricate thin films of the solution processable materials. In this method, a small amount of material is applied onto a desired substrate followed by its spreading throughout the substrate and spinning at a defined speed using a spin coater (ACT-220 DII, Saitama, Japan) as shown in Figure 1C. Optimization of film fabricated on the various surface-treated substrates was performed by varying concentration and spinning speed. The concentration range varied between 0.1 and 1 % (*w*/*v*), and the spinning speed was varied between 1000 and 3500 rpm. Finally, the OFET device was fabricated under optimized conditions of 0.5% (*w*/*v*) of RR-P3HT in dry chloroform solution at 3500 rpm for 40 s on an HMDS-treated Si/SiO_2_ substrate.

#### 2.2.2. Dip Coating

In this method, the cleaned and surface-modified substrates were vertically dipped in the desired solution, held for some time, and then lifted at a predefined speed as shown in Figure 1D, using a dip coater (Nano Dip-Coater, ND-0407, SDI, Kyoto, Japan). The concentration of RR-P3HT solution was varied from 0.05 to 1% (*w*/*v*), whereas the lifting speed was varied between 20 and 200 µms^−^^1^ for the optimization of the film quality utilizing various surface-modified substrates. After optimization, OFETs were fabricated by dip-coated RR-P3HT thin films utilizing the optimum 0.1% (*w*/*v*) concentration and 20 µms^−^^1^ of lifting speed to the HMDS-treated Si/SiO_2_ substrate.

#### 2.2.3. Unidirectional Floating Film Transfer Method

In this method, a very small amount (~10 μL) of the RR-P3HT solution in chloroform was dropped onto an orthogonal viscous liquid substrate consisting of ethylene glycol (EG) and glycerol (GL) as shown in Figure 1E. UFTM imparts not only large area uniform thin films, but also, these films are oriented perpendicular to the spreading direction of the floating film. The extent of the orientation/alignment under UFTM can be controlled by optimizing the film formation parameters such as the concentration of the polymer solution, viscosity of the liquid substrates, and temperature. In this work, efforts were directed to study the impact of different surface treatment conditions of the substrates on UFTM-processed thin films and varied the concentration of RR-P3HT polymer solution from 1.5 to 7% (*w*/*v*). Oriented thin films under optimized conditions of 4% (*w*/*v*) polymer concentration, EG: GL liquid substrate (3:1) at 50 °C on HMDS treated Si/SiO_2_ (300 nm) substrate was used for the OFET fabrication and investigation of charge transport.

### 2.3. Thin Film and Device Characterizations

#### 2.3.1. Optical and Microstructural Characterizations

The UV-vis-NIR spectrophotometer (Jasco V-570, Tokyo, Japan) was used to measure the electronic absorption spectra of the spin-coated, dip-coated, and UFTM-processed thin films. Since UFTM-processed thin films were oriented and anisotropic, to measure the extent of optical anisotropy, the Glan Thompson polarizing prism was used during the measurement of the absorption spectrum. The polarizer was positioned between the incident light source and the UFTM film, and its rotation angle controlled the polarization direction of the incident beam. This allowed for the measurement of the polarized electronic absorption spectrum, specifically in the parallel (∥) and perpendicular (⊥) directions of the film orientation. Further investigation of optical anisotropy in oriented thin films prepared by UFTM was conducted in terms of the dichroic ratio (DR), as determined by Equation (1).
(1)$$DR=\frac{Maximum\;Absorption\;at(\lambda_{max}||)}{Absorption⊥at\lambda_{max}||)}$$

For microstructural characterizations, the thin films prepared on clean Si substrates were subjected to X-ray diffraction (XRD) and grazing incidence X-ray diffraction (GIXD) measurements. The Cu-Kα radiation source-embedded with Rigaku X-ray diffractometer was utilized to perform out-of-plane XRD measurements. The X-rays incident on the sample experience a total external reflection when their index of refraction is less than unity. Additionally, the grazing incidence angle (ω) with the thin film surface is smaller than the critical angle (ω_c_) (i.e., ω < ω_c_). The sample and detector were rotated at angles of φ and 2θ_χ_, respectively, for the in-plane GIXD measurement. In addition, the angle between the film expansion during UFTM, spin coating, and dip coating of the scattering vector (χ) was fixed at approximately 0 or 90° to study anisotropy in the films. In contrast, the X-ray source and detector were rotated at angles of θ and 2θ, respectively, from the sample surface for the out-of-plane XRD measurements.

#### 2.3.2. Contact Angle Measurement

Contact angle measurements are used to study the wettability of the substrate when treated with different surface modifiers. Intermolecular interactions between the liquid and the solid surface cause the contact between them to occur, which is referred to as “wetting”, and the angle at which a liquid/vapor interface meets a solid surface is known as the contact angle, which describes the degree of wetting [25]. Hence, for the contact angle measurement firstly, the substrates were cleaned with sonication in acetone for 10 min and surface modification techniques such as UV–ozone, HMDS, OTS, and CYTOP were carried out followed by the contact angle measurement using the Kyowa Interface Science Corporation Ltd. machine (Model No: DMs-401, Saitama, Japan). For this measurement, a 10 µL droplet of DI water was dropped onto the untreated and treated substrates, and the contact angle was measured in degrees, which was later used to determine the nature of surface-treated substrates.

#### 2.3.3. Electrical Characterizations

The charge transport was investigated by fabricating OFETs in the BGTC device architecture as per the conditions described in Section 2.2. A highly doped Si substrate and a 300 nm thermally grown SiO_2_ layer were utilized as gate and gate oxide, respectively, to fabricate OFETs having an aerial capacitance of C_i_ = 10 nF cm^−^^2^. After the fabrication of OFET on to Si/SiO_2_ substrate, electrical characterizations in terms of output and transfer characteristics were performed using a computer-controlled two-channel source measurement unit (Keithley-2612, Langley, VA, USA) under 10^−^^3^ Torr.

## 3. Results

### 3.1. Contact Angle Goniometry

Several studies have shown that modifying the dielectric surface helps to improve the quality of the organic semiconductor film onto the dielectric surface, which in turn particularly enhances its electrical performance. Zan and Chou [26] reported that SAMs of surface modifiers affect the surface energy of the substrate, improving the quality of the thin films. We have used the contact angle measurement to probe the effect of various interfacial layers of the surface modifiers on the substrate. Figure 2 shows the contact angle on the treated and untreated glass substrates. The contact angle on a bare glass substrate was measured to be 54.8°, indicating its hydrophilic nature. With surface modifications of the glass substrate using HMDS, OTS, and CYTOP, the contact angle increased to 86.4°, 108.9°, and 114.7°, respectively, suggesting that the hydrophobicity of the glass surface increases in the order of CYTOP > OTS > HMDS. Lim et al. [27] reported that the contact angle with respect to the substrate increases after treatment with HMDS and OTS, indicating that the surface energy of the substrate decreases after the treatment. However, after being subjected to UV–ozone, the contact angle decreases to 36.5°, indicating an increase in the hydrophilicity of the substrate. The theoretical description of contact arises from the consideration of thermodynamic equilibrium between the three phases: the liquid phase of the droplet, the solid phase of the substrate, and the gas/vapor phase of the ambient [28]. For flat surfaces, if the surface is hydrophilic, the contact angle will be less than 90°, and the more it is towards zero degrees, the more strongly hydrophilic the solid substrate is. Moreover, if the solid surface is hydrophobic, the contact angle will be more than 90° [29]. Hence, in our case, OTS and CYTOP depict hydrophobicity, whereas HMDS shows a relatively less hydrophilic nature as compared to UV–ozone surface treatment.

### 3.2. Optimization of Thin Film Fabrication

#### 3.2.1. Spin-Coated Thin Films

Surface treatments such as HMDS, OTS, UV–ozone, and CYTOP were performed onto the cleaned glass substrate before spin coating of the RR-P3HT thin films. To understand the impact of surface modification, electronic absorption spectra were recorded (see Appendix A), and it was found that although there was a change in the peak absorbance values, there was almost no change in the positioning of the peaks, which was discernible from the normalized absorption spectra as shown in Figure 3A. It can be seen from this figure that there was the presence of a dominant peak at 550 nm (A_0-1_) corresponding to interchain interactions between the polymeric chains of the RR-P3HT. At the same time, two shoulders one of lower energy appearing at 500 nm (A_0-2_) and another at 600 nm (A_0-0_) associated with the π-π* electronic transition aggregation behavior of the polymeric chains [30,31]. It is worth mentioning that it was not possible to spin coat an RR-P3HT thin film on a CYTOP-treated glass substrate owing to its extremely high hydrophobic nature (Appendix A). Considering the normalized absorption spectra of RR-P3HT on HMDS and OTS-treated glass substrates, which are almost similar to (Figure 1A) and uniform with the fabricated thin films (Appendix A), it could be concluded that both of the surface treatments are suitable for spin coating. The spinning speed and concentration of the solution under spin coating play a dominant role in controlling the uniformity and morphology of the fabricated thin films. Spinning speed is an important factor, which not only affects the thickness but also the quality of the fabricated thin films. Therefore, it is important to optimize the effects of spinning speed, which varied from 1000 to 3500 rpm, keeping other variable parameters such as the nature of the substrate surface, and polymer concentration to be fixed. Electronic absorption spectra as shown in Figure 3B reveal that the thickness of the film increases by decreasing the spinning speed, and there was about 1.5 times increase in the thickness of RR-P3HT thin films upon a decrease in the spinning speed from 3500 rpm (32 nm), 3000 rpm (36 nm), 2000 rpm (39 nm) to 1000 rpm (47 nm) [27]. In the present work, the spinning speed of 3500 rpm was used for OFET fabrication since thinner films are preferred for planar charge transport because the uniformity and consistency of the film begin to be affected as we move towards lower spinning speeds [27].

For optimizing the effect of concentration, the concentration of the polymer solution was varied from 0.1 to 1% (*w*/*v*) while keeping the UV–ozone treated glass substrate and spinning speed of 3500 rpm as fixed parameters. It can be seen from the absorption spectra (Appendix A) that the thickness of the film increased on increasing the concentration of the solution, and it was also validated that spin-coated films are isotropic since there was no color contrast in fabricated thin films upon rotating the polarizer from 0° to 90° (Appendix A). In the normalized absorption spectra (Figure 3C), the A_0-0_, A_0-1_, and A_0-2_ transition peak values for the spin-coated thin film have been compared. These transition peaks refer to the absorption peaks and shoulders associated with the different electronic and vibrational transitions within the material. A_0-0_ transition peak for RR-P3HT correlates to the intermolecular interactions and presence/absence of defects. The A_0-1_ transition peak is associated with the π-π* transition. This transition is influenced by the degree of overlap between adjacent polymer chains, known as π-π stacking. The intensity and energy of the A_0-1_ peak are sensitive to factors such as the degree of polymer crystallinity, chain orientation, and π-π interactions. A_0-2_ correlates to the π-electron conjugation along the polymer backbone (intrachain interactions), reflecting the structural fluctuations or conformational changes along the polymer backbone [32,33]. Spano thoroughly looks into how A_0–0_ modes grow in RR-P3HT, connecting them closely with the way the electronic structure of excitonic bandwidth (W) and intermolecular coupling transition energy (E_P_) relate to each other, as described by Equation (2) given below.
(2)$$\frac{A_{0-0}}{A_{0-1}}={\left\{\frac{1-0.24\frac{W}{Ep}}{1+0.073\frac{W}{Ep}}\right\}}^{2}$$

The ratio A_0-0_/A_0-1_, which is related to the coupling energy (W/E_P_) (for Huang–Rhys Factor = 1; W = Exciton bandwidth), provides information about the strength of electronic interchain coupling and degree of vibrational relaxation (E_P_) across various polymer chains in the thin film. A higher value of A_0-0_/A_0-1_ or a lower value of W/E_P_ is typically associated with an increase in interchain ordering and crystalline domain arrangement, where electronic excitations can propagate with minimal energy dissipation through vibrational relaxation [30]; (A_0-0_/A_0-1_) for 0.8% and 1% solution is higher in comparison to other concentration of solution in their thin film state as given in Table 1. This shows that 0.8% and 1% solution-based thin films are highly ordered and crystalline with respect to the intermolecular interactions. On the other hand, the ratio of A_0-2_/A_0-1_, which corresponds to the intrachain coupling, is lower for 0.1 and 0.2% concentration as compared to the other concentrations. A lower A_0-2_/A_0-1_ (Table 1) gives a lower exciton bandwidth resulting in disordered and narrowly distributed conjugation length. A concentration of 0.5% (*w*/*v*) for spin-coated thin films gives a good balance in attaining the coupling interactions for intramolecular as well as intermolecular ordering and hence, the 0.5% concentration was taken for the fabrication of OFETs for electrical characterizations.

#### 3.2.2. Dip-Coated Thin Films

Dip coating for thin film fabrication was developed to prepare large-area uniform thin films, and for CPs, it offers the advantage of utilizing their inherent molecular self-assembly owing to extended π-conjugation. At the same time, it is also advantageous over spin coating in terms of material utilization. Parametric optimization such as the nature of the substrate surface, polymer concentration, and lifting speeds has to be performed to control the film uniformity and morphology. Electronic absorption spectra dip-coated thin films prepared on differently surface-modified glass substrates are shown in the Appendix A (Appendix A) while keeping the polymer concentration and lifting speed of 1% (*w*/*v*) and 20 µm/s, respectively. It was observed that dip-coated films have three distinct peaks A_0-0_, A_0-1_, and A_0-2_ occurring at wavelengths of 600 nm, 550 nm, and 510~520 nm, respectively, as shown in Figure 4A, and the respective values are given in Table 2. It has been widely reported that in RR-P3HT when the ratio of peak absorbances corresponding to A_0-0_/A_0-1_ is higher, it signifies enhanced crystallinity in the followed by HMDS suggesting slightly better molecular ordering in the thin films fabricated on OTS [34]. Despite this better ordering in the OTS-treated glass substrate, the film uniformity is inferior to that on the treated substrate, which can be seen from the photographic image shown in the Appendix A (Appendix A). Moreover, dip-coated films could not be fabricated onto CYTOP-treated surfaces (Appendix A) due to the highly hydrophobic nature of CYTOP-treated substrates. Hence, for the dip-coating technique, HMDS surface treatment would be the most suitable as it results in a homogenous and uniform film without seriously affecting the molecular ordering in the fabricated thin film.

For the optimization of the effect of the concentration of CPs in dip-coating techniques, optical characterizations were performed on the glass substrates varying the concentration from 0.05 to 1% (*w*/*v*) and keeping the other variable parameters such as UV–ozone-treated glass substrates at the lifting speed of 20 μm/s. A perusal of Appendix A reveals that there was about a six times increase in the thickness upon increasing the polymer concentration from 0.05% to 1.0% (*w*/*v*). Further, from normalized absorption spectra shown in Figure 4B, we confirm that the ratio A_0-0_/A_0-1_ decreases with the increasing polymer concentration and was highest for the 0.1% (Table 2) concentration, suggesting that the molecular ordering and crystallinity are the best in this case. Moreover, when we compare the ratio of A_0-2_/A_0-1_, it can be concluded that the 0.1% concentration exhibited the smallest molecular disordering, and therefore, the 0.1% concentration was found to be optimum and was used for the OFET device fabrication. Further, we also evaluated the W for different concentrations of the CP solution, suggesting that the 0.1% CP concentration has the lowest value of W, thus the highest crystallinity. Lifting speed during dip coating is one of the most important parameters, which not only controls the thickness but also the quality (uniformity and morphology) of the fabricated thin films. Optimization of lifting speed in the dip coating was performed by varying the lifting speed from 10 to 200 µm/s, keeping the other variable parameters such as the polymer concentration and nature of the substrate to 0.2% (*w*/*v*) and UV–ozone-treated glass substrate, respectively, to be fixed, and the results are shown in Appendix A. It can be seen from this figure shown in the Appendix A that by decreasing the lifting speed from 10 µm/s to 200 µm/s, there was a >10 times increase in film thickness (peak absorbance). At the same time, normalized absorption spectra shown in Figure 4C reveal that the ratio of the A_0-0_/A_0-1_ peak for 20 µm/s and 10 µm/s lifting speed is almost similar, pointing towards a more ordered and crystalline film as compared to other lifting speeds. However, the films prepared at 10 µm/s speed were too thin and non-uniform, and therefore, 20 µm/s was found to be the optimum for the device fabrication. Moreover, it was observed that dip-coated thin films are isotropic and non-oriented owing to the same color of the film (no contrast) under the parallel and perpendicular polarizer (Appendix A).

#### 3.2.3. UFTM Fabricated Thin Films

The uniqueness of UFTM lies in the fact that we can not only fabricate a large area and uniform thin films but also that they are anisotropic or oriented. At the same time, the least material wastage and multilayer film fabrication by a solution-based approach are its added advantages. The extent of the molecular orientation under UFTM can be controlled by controlling the film fabrication parameters like polymer concentration, temperature, and viscosity of the liquid substrate [35]. It has been reported by our group previously that for RR-P3HT, a liquid substrate consisting of EG and GL in a 3:1 ratio gives the highest orientation, which has been used in the present work for the optimization of UFTM-processed thin films. To study the impact of surface treatment on UFTM-processed thin films, thin films were fabricated onto the HMDS, OTS, UV–ozone, and CYTOP-treated glass substrates keeping the other parameters like EG:GL (3:1) liquid substrate and 4% of the polymer concentration fixed. The uniformity of the film and the presence of orientation can be clearly seen in the photographic images taken in the absence and presence of a polarizer film (see Appendix A) [36]. Polarized electronic absorption spectra were measured for UFTM-processed thin films fabricated on differently surface-modified glass substrates, which are shown in Figure 5A. Optical anisotropy in terms of the optical dichroic ratio (DR) was calculated from the polarized absorption spectra and shown in Table 1. It can be seen from this figure and table that the nature of the surface treatment does not seriously affect the orientation of RR-P3HT, which was slightly better on CYTOP-modified glass substrate with a DR of 4.3. A perusal of the polarized absorption spectra reveals that in thin films of RR-P3HT fabricated by UFTM, there is a clear presence of absorption peaks around 550 nm associated with A_0-1_ along with vibronic shoulders associated with A_0-0_, and A_0-2_. The pronounced absorption intensity for the A_0-1_ peak as compared to the A_0-0_ shoulder corroborates that UFTM-processed films are more crystalline and ordered owing to enhanced inter-chain interactions. From the normalized absorption spectra (Appendix A), it can be observed that the ratio of A_0-0_/A_0-1_ associated with excitonic coupling energy (W) is almost similar and higher in the case of CYTOP and HMDS surface treatments pointing to more ordered and crystalline films as compared to UV–ozone and OTS-treated substrates. However, the transition peak of A_0-2_ is more dominant in the case of the HMDS-treated substrate in comparison to CYTOP. Higher DR and better film quality make CYTOP one of the most suitable surface treatments for UFTM; this may be due to the higher hydrophobicity of CYTOP, which aids in easily casting the solid film from a viscous liquid substrate.

The concentration of the polymer solution under UFTM plays a dominant role in not only controlling the film thickness but also molecular ordering and orientation by controlling the spreading speed of the polymer solution on the orthogonal viscous liquid substrate. In this present work, the effect of the polymer concentration on the nature of the fabricated thin film was estimated by varying the concentration of the RR-P3HT solution from 1.5 to 7% (*w*/*v*) while keeping other variables like the EG:GL (3:1) liquid substrate and UV–ozone-treated glass substrate as fixed parameters. Polarized absorption spectra recorded for oriented thin films fabricated using varying polymer concentrations are shown in Figure 5B along with the summary of calculated optical anisotropy in terms of optical DR in Table 3. It can be seen from this figure and table that there is about a four times increase in the thickness of UFTM-processed thin films upon increasing the polymer concentration from 1.5% to 7.0%, which is almost linear.

Moreover, the dichroic ratio calculated for various concentrations as summarized in Table 1 indicates a gradual increase followed by a decrease in DR, with a maximum DR at the 4% polymer concentration suggesting the presence of maximum orientation. From the normalized spectra in Appendix A, it was observed that the ratio of the vibronic peak of A_0-0_/A_0-1_ for lower concentrations is better in comparison to higher concentrations of films, which also have dominant A_0-2_ peaks. Moreover, there is an observed redshift for lower concentrations of the CP solution, indicating that lower concentration films were more ordered and crystalline. However, considering the better DR, good film quality and optimum thickness for ease of fabrication and uniformity, a concentration of 4% was found to be optimum and was used for OFET fabrication. Moreover, the crystallinity for UFTM films (102 meV) is much higher than the spin-coated films but lower than the dip-coated films.

### 3.3. Microstructural Characterization

To explore the crystallinity and macromolecular orientation, in-plane GIXD and out-of-plane XRD measurements were performed for the fabricated thin films of the RR-P3HT (Figure 6). From the out-of-plane XRD analysis of RR-P3HT films fabricated via UFTM and dip coating as shown in Figure 6A, a sharp diffraction peak corresponding to the lamellar stacking of the alkyl side chains up to the 3rd order, at θ values of 5.5°, 11°, and 16°, which corresponds to the (hkl) value of (100), (200), and (300), respectively, was evident. In addition to this, the thin film of UFTM also demonstrated a 2θ value of 23.5° corresponding to (010), with a plane separation value of 3.9 Å (b or c-axis, lattice constant), and lattice separation at 2θ of 5.5° is 16 Å corresponding to the lattice constant of a-axis of RR-P3HT [37,38]. From this analysis, it can be stated that in RR-P3HT, the polymer chain exhibits an orientation close to parallel alignment with the substrate, while the individual thiophene rings within the polymer structure are oriented perpendicular to the substrate surface; these observations are in close relation with the previously reported works [39]. Moreover, the spin-coated thin film exhibits a comparatively weak (100) peak of the second order, indicating that the films made through dip coating and UFTM are more crystalline. Also, it is depicted from these diffraction peaks that RR-P3HT films fabricated via UFTM are more crystalline than its contrary dip coating film, as its (100) peak is not only weaker, but also, its full width at half maximum is relatively narrow. Further, from the analysis of in-plane GIXD of RR-P3HT films shown in Figure 6B, it was clear that the films prepared using UFTM and dip coating did not display any diffraction peaks linked to alkyl side chains, as it only showcased a diffraction peak, with an (hkl) value of (010), at ~23.3° in both the cases, which reveals that all crystallites are oriented edge-on [23]. Apart from this, the spin-coated film of the CP exhibited no (010) diffraction peaks, because this film has its crystallite fractions, oriented face-on [37]. Thus, the prepared RR-P3HT films by dip coating and UFTM are more ordered and crystalline than spin-coated films, which is attributed to the results of the in-plane and out-of-plane XRD profiles of the films. While comparing the results of this study with other pre-existing literature, it was observed that Yang et al. [39] reported P3HT/CHCl3 films, mostly have a face-on crystal orientation that is highly favorable toward the vertical charge transport for devices like diodes and solar cells. In another study by De Long Champ et al. [40], it was reported that spin-coated RR-P3HT/CHCl_3_ thin films at a spinning speed of ~2000 rpm was preferentially face-on oriented. Hence, from this analysis, it was revealed that the spin-coated films are in majority face-on and UFTM and dip-coating films were oriented edge-on.

### 3.4. Electrical Characterization

After the initial studies to investigate the effect of crystallinity and molecular orientation of CP thin films of RR-P3HTs fabricated using UFTM, dip-coating, and spin-coating; OFETs were fabricated in the BGTC device architecture as shown in Figure 1B, for investigating anisotropic charge transport. In the previous reports by our group, it has already been reported that thin film fabrication by UFTM is oriented edge-on, which facilitates the facile in-plane charge transport [41]. In addition to this, Xue et al. [42] reported that RR-P3HT dip-coated films are edge-on oriented. A spin-coated RR-P3HT film as a semiconductor demonstrates face-on orientation or mixed orientation. In this study for OFET fabrication, initially, Si/SiO_2_ substrates were surface-treated with HMDS to make the substrate’s surface hydrophobic before thin film fabrication using RR-P3HT because the polymer under investigation is hydrophobic. Then, the thin film was fabricated by using optimum concentrations of RR-P3HT solution for different thin film fabrication techniques, which are 0.5%, 0.1%, and 4% (*w*/*v*) for spin coating, dip coating, and UFTM, respectively. The SiO_2_ gate oxide dielectric thickness was kept constant at 300 nm. Finally, from the output and transfer characteristics obtained by the I–V (current–voltage) characteristics shown in Figure 7, it was evident that the output curves depicted by Figure 7i demonstrated p-type semiconducting behavior, as evident from the modulation of channel current upon application of the differential negative bias voltage. In addition, Equation (3) was used to obtain the field effect mobility (μ) from the transfer curves shown in Figure 7ii in the saturation region, where μ, C_i_, W, and L stand for saturated mobility, areal capacitance, channel width, and channel length, respectively.
(3)$$I_{Ds}=\frac{µC_{i}W}{2L}{\left(V_{GS}-V_{DS}\right)}^{2}$$

Source-drain current (I_DS_), gate voltage (V_GS_), and drain-source voltage (V_DS_) are the corresponding terms. In the saturated region, the slope of the |I_DS_|^1/2^ versus V_GS_ plot was used to calculate the μ, while the ratio of the on-to-off current in the transfer characteristic was used to calculate the on-off ratio [11]. Further, for the fabrication of OFETs by UFTM-processed oriented thin films, parallel and perpendicular OFETs were also fabricated by aligning the source and drain electrodes of the channel towards both the parallel and perpendicular to the orientation direction to study anisotropic charge transport. The output and transfer characteristics of the fabricated OFETs by spin coating (Figure 7A), dip coating (Figure 7B), and UFTM (Figure 7C,D) are presented in Figure 7.

The calculated value of μ and on/off ratio from the transfer characteristics for the fabricated OFETs by utilizing RR-P3HT thin films prepared by spin coating, dip coating, and UFTM are summarized in Table 4.

A perusal of Figure 7 and Table 1 reveals that in OFETs fabricated using spin-coated RR-P3HT, there was a significant leakage in current at zero gate bias, which resulted in an extremely low on–off ratio of 3.6 × 10^2^ and μ of 8.0 × 10^−4^ cm^2^V^−1^s^−1^. On the other hand, for µ estimated for the dip-coat-fabricated OFET, there was about two times improvement in the carrier mobility (1.3 × 10^−3^ cm^2^V^−1^s^−1^), while there was an order of magnitude enhancement in the on/off ratio (4.9 × 10^3^) as compared to its spin-coated OFETs device counterparts. Interestingly, OFETs that used parallel-oriented RR-P3HT thin films showed significantly improved charge carrier transport, with µ and on/off ratios of 7.0 × 10^−2^ cm^2^V^−1^s^−1^ and 4.5 × 10^4^, respectively. Apart from this, OFETs that were fabricated using oriented thin films by UFTM demonstrated an electrical anisotropy (µ^ǁ/^μ^⊥^) of 2. When compared to their UFTM and dip-coating-processed device counterparts, the presence of face-on fractions in spin-coated thin films with alkyl chains positioned in the substrate plane results in reduced in-plane charge transport and impeded OFET performance. In the dip-coating technique, fabricated OFETs exhibited lower mobility than the UFTM-processed device, due to the absence of orientation and the presence of relatively higher disordered polymer domains in the film. OFETs fabricated using UFTM showed an enhanced planar charge transport, with an average μ that was more than two orders of magnitude higher (7.0 × 10^−2^ cm^2^V^−1^s^−1^) than that of OFET spin-coated thin films (8 × 10^−4^ cm^2^V^−1^s^−1^). Enhanced molecular orientation and crystallinity resulted in a notable improvement in OFETs with an active layer made using UFTM that is nearly 10 times higher μ value compared to RR-P3HT films prepared by dip coating and nearly 100 times higher by spin coating. Since the crystallinity of dip-coated and UFTM-processed thin films are almost similar but about 50 enhancements in the carrier mobility of UFTM-processed thin films suggest that molecular orientation undoubtedly plays a dominant role as compared to crystallinity in controlling the charge transport.

## 4. Conclusions

In this study, thin films of the RR-P3HT have been fabricated using three different solution-processable thin film fabrication techniques, namely spin coating, dip coating, and UFTM. Initially, optimizing various parameters affecting the thin film properties was carried out in detail using optical and microstructural characterizations followed by the fabrication of OFETs. Thin films fabricated by UFTM were large-area, oriented, crystalline, and anisotropic. In contrast, spin-coated and dip-coated thin films were non-oriented and isotropic, but dip-coated films were more crystalline as compared to spin-coated films, which was confirmed by the electronic absorption spectra and XRD results. Moreover, various parameters affecting the quality of the thin film were studied in detail, such as surface treatment, and it was found that a strongly hydrophobic surface was better for UFTM, whereas mild hydrophobicity of the surface ensured the best film quality in the case of spin coating and dip coating. Further, the XRD and GIXD results show that macromolecular chains in UFTM and dip-coated films were edge-on oriented facilitating better planar charge transport as compared to spin-coated films. Finally, OFETs were fabricated to analyze the implications of molecular orientation and crystallinity on device performance. Thus, the electrical parameters such as mobility verified that edge-on orientation and crystallinity enhance the device performance in OFETs fabricated by UFTM and dip coating when compared to spin coating. There was a notable enhancement (>10^2^ times) in mobility for UFTM-processed thin films of RR-P3HT as compared to spin-coated device counterparts, which was attributed to the enhancement of crystallinity and the presence of the molecular orientation, Hence, this study underscores the promising prospects of utilizing UFTM and dip coating for fabricating high-performance OFETs with enhanced charge mobility and on/off ratios compared to spin coating.

## Figures and Tables

**Figure 1 micromachines-15-00677-f001:**
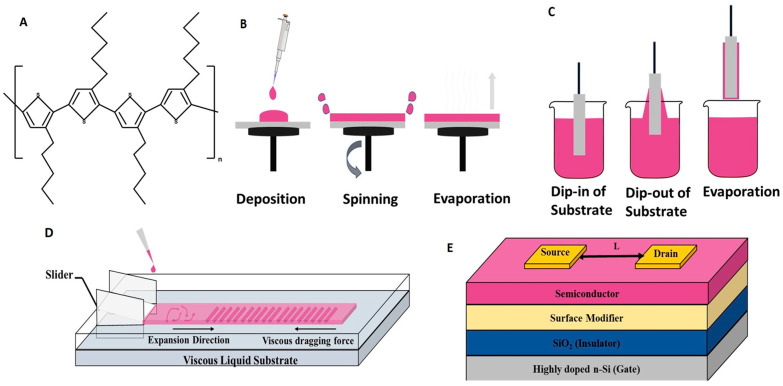
(**A**) Structure of RR-P3HT, (**B**) spin coating, (**C**) dip coating, (**D**) UFTM, and (**E**) schematic representations for OFET architecture.

**Figure 2 micromachines-15-00677-f002:**
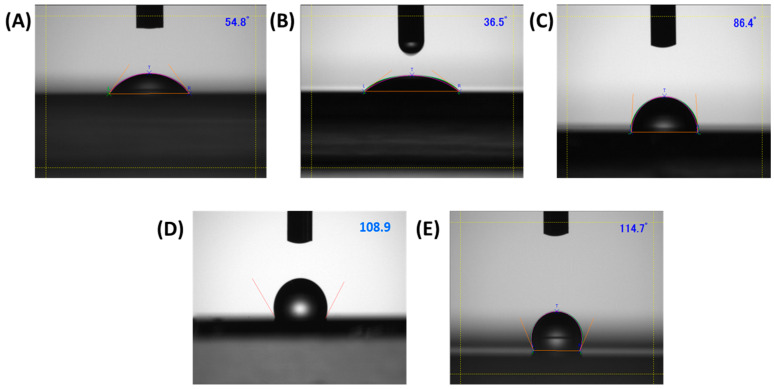
Contact angle of the glass substrate surface (**A**) untreated, (**B**) UV–ozone treated, (**C**) HMDS, (**D**) OTS, and (**E**) CYTOP.

**Figure 3 micromachines-15-00677-f003:**
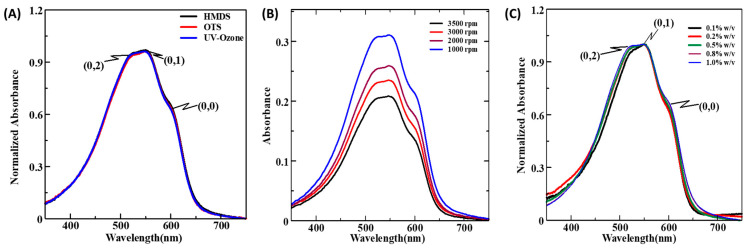
(**A**) Normalized absorption spectra for spin-coated RR-P3HT thin films on glass substrate treated with different surface modifiers, (**B**) absorption spectra at various spinning speeds, and (**C**) normalized absorption spectra of the films fabricated with varying polymer concentrations in chloroform.

**Figure 4 micromachines-15-00677-f004:**
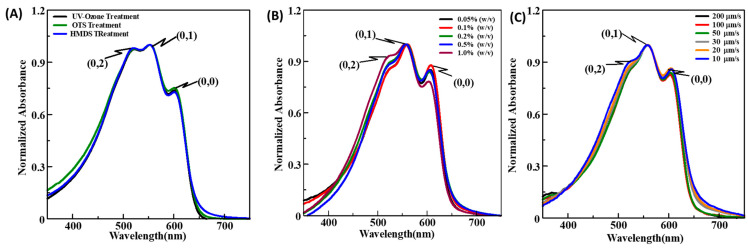
Normalized absorption spectra for dip-coated RR-P3HT thin films: (**A**) on the differently surface-modified glass substrates, (**B**) at different concentrations of the polymer, and (**C**) at various lifting speeds.

**Figure 5 micromachines-15-00677-f005:**
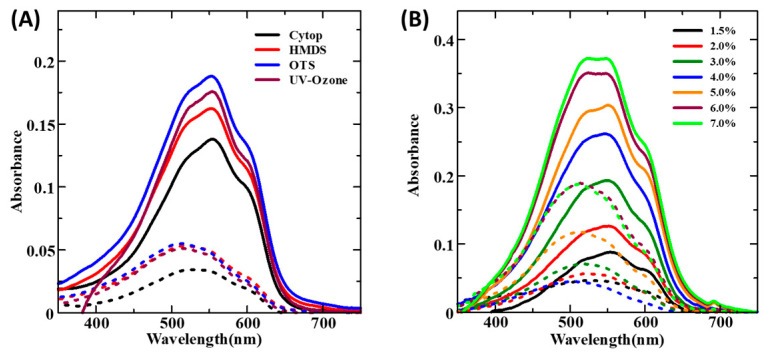
Polarized electronic absorption spectra for UFTM-processed RR-P3HT thin films: (**A**) on various surface treatments, and (**B**) at different concentrations. Solid and dotted lines represent spectra taken under parallel and perpendicular polarization.

**Figure 6 micromachines-15-00677-f006:**
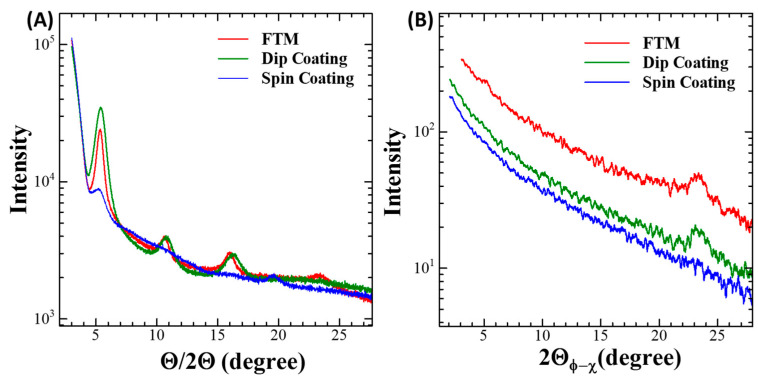
(**A**) Out-of-plane XRD, and (**B**) in-plane GIXD profiles for the thin films fabricated with UFTM, dip coating, and spin coating.

**Figure 7 micromachines-15-00677-f007:**
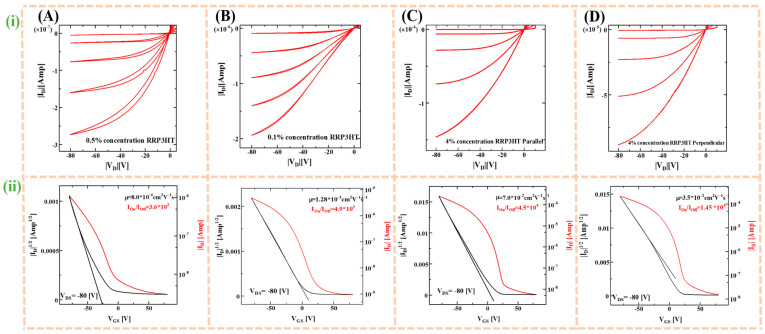
(**i**) Output and (**ii**) transfer characteristics of (**A**) spin-coated, (**B**) dip-coated, (**C**) UFTM-processed parallel, and (**D**) UFTM-processed perpendicular OFETs.

**Table 1 micromachines-15-00677-t001:** Optical parameters of spin-coated thin films.

CP Conc. (*w*/*v*)	A_0-0_	A_0-1_	A_0-2_	A_0-0_/A_0-1_	A_0-1_/A_0-2_	Exciton Bandwidth (meV)
0.1%	0.033	0.053	0.050	0.624	0.655	127
0.2%	0.060	0.097	0.094	0.620	0.636	128
0.5%	0.129	0.200	0.195	0.647	0.665	118
0.8%	0.192	0.288	0.284	0.667	0.676	110
1.0%	0.217	0.322	0.318	0.674	0.682	108

**Table 2 micromachines-15-00677-t002:** Optical parameters of dip-coated thin films.

CP Conc. (*w*/*v*)	A_0-0_	A_0-1_	A_0-2_	A_0-0_/A_0-1_	A_0-1_/A_0-2_	Exciton Bandwidth (meV)
0.05%	0.089	0.105	0.089	0.850	0.999	45
0.1%	0.166	0.189	0.161	0.878	1.035	36
0.2%	0.231	0.276	0.245	0.837	0.943	49
0.5%	0.356	0.420	0.369	0.846	0.964	46
1.0%	0.453	0.579	0.538	0.782	0.841	68

**Table 3 micromachines-15-00677-t003:** Effect of surface modification of glass and polymer concentration on the optical anisotropy of the UFTM processed thin films.

Polymer Conc. (*w*/*v*)	UV-Ozone	HMDS	OTS	CYTOP	Exciton Bandwidth (meV)
1.5%	1.8				
2.0%	2.5				
3.0%	3.1				
4.0%	4.2	4.1	4.0	4.3	102
5.0%	3.5				
6.0%	2.3				
7.0%	2.1				

**Table 4 micromachines-15-00677-t004:** Electrical parameters of OFETs fabricated using RR-P3HT thin films fabricated by various techniques such as spin coating, dip coating and UFTM.

Fabrication Techniques	Concentration (*w*/*v*)	Thickness (nm)	Carrier Mobility (cm^2^V^−1^s^−1^)	On-Off Ratio
Spin coating	0.5%	34	8.0 × 10^−4^	3.6 × 10^2^
Dip coating	0.1%	29	1.3 × 10^−3^	4.9 × 10^3^
UFTM	4.0%	23	7.0 × 10^−2^	4.5 × 10^4^

## Data Availability

Data will be made available on request.

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
