# Peer review of "Clarifying the Dominant Role of Crystallinity and Molecular Orientation in Differently Processed Thin Films of Regioregular Poly(3-hexylthiophene)"

_micromachines, 2024, doi:10.3390/mi15060677_

Round 1

Reviewer 1 Report

Comments and Suggestions for Authors

In this paper, Pandey and co-authors have demonstrated crystallinity and molecular orientation can be largely affected by different film processing technologies. Specifically, they chose P3HT, a classic organic semiconducting polymer, as an example, and processed it by three film preparation methods, i.e., spin-coating, dip-coating, and unidirectional 13 floating-film transfer method (UFTM). I recommend this paper could be accepted after addressing the following concerns.

1.    The molecular weight of P3HT used in this paper should be specified.

2.    The description of Figure S1D is missing in Figure S1 caption.

3.    Figure 2 caption is unclear, ‘treated’ has been used twice and are the order numbers before or after the description? ” A) untreated, and treated UV-Ozone 250 treated (B), HMDS (C),…”

4.    In the 3.2.1.Spin-coated thin films section, for Figure 3C, it is hard to see any differences in their spectra. As the ratios A0-0/A0-1 and A0-2/A0-1 are applied to justify the 0.5%(w/v) as the ideal concentration, can you specify these ratios in numbers for a better understanding of the differences?

5.    Same for the ratios A0-0/A0-1 and A0-2/A0-1 for Figure 4. Is it better to add a table?

Comments on the Quality of English Language

The English is okay to understand and no obvious grammar errors. Academic writing can be improved.

Author Response

Reviewer 1

In this paper, Pandey and co-authors have demonstrated crystallinity and molecular orientation can be largely affected by different film processing technologies. Specifically, they chose P3HT, a classic organic semiconducting polymer, as an example, and processed it by three film preparation methods, i.e., spin-coating, dip-coating, and unidirectional floating-film transfer method (UFTM). I recommend this paper could be accepted after addressing the following concerns.

Our response: Authors are thankful for the comments and suggestions of the reviewer

and will try our best to confer with the comments and suggestions of the reviewer along with

their suitable incorporation in the revised manuscript.

Reviewer 1, Comment 1: The molecular weight of P3HT used in this paper should be specified.

Our response: The authors are thankful to the reviewer. We have added the Molecular weight and PDI values in line 101 of page 3 in the revised manuscript.

Reviewer 1, Comment 2: The description of Figure S1D is missing in Figure S1 caption.

Our response: Thank you very much for this comment and we are sorry for this mistake. We have added the figure description in S1 D in the revised supplementary data.

Reviewer 1, Comment 3: Figure 2 caption is unclear, ‘treated’ has been used twice and are the order numbers before or after the description? ”A) untreated, and treated UV-Ozone 250 treated (B), HMDS (C),…”

Our response: Thank you very much for this comment. We have added the description of Figure 2(E) in lines 253-254 in the revised manuscript.

Reviewer 1, Comment 4: In the 3.2.1. Spin-coated thin films section, for Figure 3C, it is hard to see any differences in their spectra. As the ratios A0-0/A0-1 and A0-2/A0-1 are applied to justify the 0.5%(w/v) as the ideal concentration, can you specify these ratios in numbers for a better understanding of the differences?

Our response: We agree with the concerns of the reviewer and have provided a detailed explanation for the ratios A0-0/A0-1 and A0-2/A0-1 in Table 1 of the revised manuscript.

Reviewer 1, Comment 5:  Same for the ratios A0-0/A0-1 and A0-2/A0-1 for Figure 4. Is it better to add a table?

Our response: Thank you very much for this comment and a similar concern has also been raised by the reviewer in the previous question. We have provided a detailed explanation about the ratios A0-0/A0-1 and A0-2/A0-1 are applied to justify the 0.1 %(w/v) as the ideal concentration in Table 2 in the revised manuscript.

Reviewer 2 Report

Comments and Suggestions for Authors

Printed organic electronics is an emerging technology that attract considerable interest in recent years as it enables the fabrication of large-scale, low-cost electronic devices, and thus offers significant possibilities in terms of developing new applications in various fields. Easy processing is a prerequisite for the development of low-cost, flexible and printed electronic devices. Among processing techniques, dip-coating, and spin coating methods are considered simple, efficient, and low-cost methods to fabricate electronic devices. One of the major challenges is the control of thin film morphology, molecular orientations, crystallinity of thin film, and directional alignment of polymer films during coating processes. This work investigates the crystallinity and molecular orientation of  regio-regular poly(3-hexylthiophene)(P3HT) to fabricate OFETs, where thin films of P3HT were fabricated by different methods: UFTM, dip-coating, and spin coating. Various parameters were first optimized under each type of film fabrication method followed by their optical and microstructural characterizations. Finally, OFETs were fabricated to analyse the implications of molecular orientation, uniformity and crystallinity on the charge transport, and device performance. Overall this work is of interest for organic and printed electronic communities.

Below are my comments.

 Minor comments.

Line 48 to 50 " electronic devices, organic semiconductors is the pillar. This sentence is is incorrect, please re-formulate it.

Line 54 " For practical applications, managing the cost and perfor-54 mance of these OFETs are the primary challenges" Please reformulate this sentence as "One of the challenges faced by organic electronic devices is managing the cost and performance"

Line 57 " OFETs are planar devices, a homogenous and aligned thin film in the direction of the channel with an edge-on orientation is preferred for the better device performance." please re-write as: " homogenous thin film with aligned molecules along the channel direction.

Line 59 the authors stated "There are several methods for thin film fabrication, such as bar coating[15], mechanical rubbing[16], and friction, etc.

There are many  processing techniques for OSc, such meniscus guided coating methods which are considered simple, efficient, and low-cost methods to fabricate electronic devices in industry. For example , see  review article , https://doi.org/10.1016/j.cis.2019.102080

In Figure 1 Please delete numbering (2, 5)  in the chemical structure of RR-P3HT. Please improve the schematic representation of dip coating and spin coating.

line 126 Superdehydrated hexane. please use dry hexane

line 142 "patterning source/drain electrodes using thermal evaporation" Please provide the thickness of Au source/drain

line 152 Please change "super dehydrated chloroform" by " dry chloroform"

line 227 please change to  " improve the quality of the organic semiconductor film onto the dielectric surface which in turn particularly enhances its electrical performance"

Line 261 " chains of the P3HT" please keep the same abbreviation RR-P3HT.

Line 265 P3HT change to RR-P3HT

Line 266 please delete this " (Fig. S1C. Please remove Figure S1B)."

Line 276 the authors claimed that " the thickness of the film increases by decreasing the spinning speed and there was about 1.5 times increase" Can the authors provide a value of the effective thickness of the film for each speed?

 Major comments

1) The authors attempt a quantitative analysis of crystallinity of RR-P3HT thin film from UV-vis absorption spectra of thin film fabricated by three different methods. The P3HT crystallinity can be obtained by comparing the two vibronic peaks at 555 nm and 610 nm using a Spano analysis. (Spano et al. Phys. Rev. Lett. 2007, 98.).  please add more detail in this analysis. In such model the ratio of the two H-aggreagate absorbance peaks is given by [(1 − 0.24W /Ep)/(1 + 0.73W /Ep)]2, where Ep = 0.18 eV is the energy of the symmetric C=C stretch, and W is the free exciton bandwidth within the crystalline phase. The ratio of the two vibronic peaks is related the crystallinity phase in the film.

2) It is widely observed that coating P3HT using low speed in meniscus guided coating methods, such as deep-coating, yields to patterned films. (https://doi.org/10.1016/j.cis.2019.102080). The authors have used very low lifting speed, 20 to 200 μms-1 for the fabrication of P3HT film. My question is:  Do the authors check the thin film morphology using optical or AFM microscopies?

Comments on the Quality of English Language

It would be nice if the authors can improve the quality of English and verfy typos, etc.

Author Response

Reviewer 2

Reviewer 2, Comment Summary:

Printed organic electronics is an emerging technology that attract considerable interest in recent years as it enables the fabrication of large-scale, low-cost electronic devices, and thus offers significant possibilities in terms of developing new applications in various fields. Easy processing is a prerequisite for the development of low-cost, flexible and printed electronic devices. Among processing techniques, dip-coating, and spin-coating methods are considered simple, efficient, and low-cost methods to fabricate electronic devices. One of the major challenges is the control of thin film morphology, molecular orientations, crystallinity of thin film, and directional alignment of polymer films during coating processes. This work investigates the crystallinity and molecular orientation of regioregular poly(3-hexylthiophene) (P3HT) to fabricate OFETs, where thin films of P3HT were fabricated by different methods: UFTM, dip-coating, and spin coating. Various parameters were first optimized under each type of film fabrication method followed by their optical and microstructural characterizations. Finally, OFETs were fabricated to analyse the implications of molecular orientation, uniformity and crystallinity on the charge transport, and device performance. Overall this work is of interest to organic and printed electronic communities.

Our response: Authors are thankful for the insightful comments, encouragement and suggestions of the reviewer and we have tried our best to confer with the comments and suggestions of the reviewer along with their suitable incorporation in the revised manuscript.

Below are my comments.

Minor comments.

Reviewer 2, Comment 1: Line 48 to 50 " electronic devices, organic semiconductors is the pillar. This sentence is incorrect, please re-formulate it.

Our response: We are thankful to the reviewer for this comment. We have reformulated the sentence in lines 48-50 of page 2 in the revised manuscript.

Reviewer 2, Comment 2: Line 54 " For practical applications, managing the cost and performance of these OFETs are the primary challenges" Please reformulate this sentence as "One of the challenges faced by organic electronic devices is managing the cost and performance"

Our response: We are thankful to the reviewer for this suggestion We have reformulated the sentence in lines 54-55 of page 2 in the revised manuscript.

Reviewer 2, Comment 3: Line 57 " OFETs are planar devices, a homogenous and aligned thin film in the direction of the channel with an edge-on orientation is preferred for the better device performance." please re-write as: " homogenous thin film with aligned molecules along the channel direction.

Our response: We are thankful to the reviewer for this suggestion. We have reformulated the sentence in lines 57-58 in the revised manuscript.

Reviewer 2, Comment 4: In line 59 the authors stated "There are several methods for thin film fabrication, such as bar coating [15], mechanical rubbing [16], and friction, etc.

There are many processing techniques for OSC, such as meniscus-guided coating methods which are considered simple, efficient, and low-cost methods to fabricate electronic devices in industry. For example, see the review article, https://doi.org/10.1016/j.cis.2019.102080

Our response: Thank you very much for this valuable comment and suggestion. We have added the necessary suggestion of the reviewer in lines 59-66 and added the reference as suggested by the reviewer in the revised manuscript.

Reviewer 2, Comment 5: In Figure 1 Please delete numbering (2, 5) in the chemical structure of RR-P3HT. Please improve the schematic representation of dip coating and spin coating.

Our response: Thank you very much for this suggestion. We have changed the chemical structure of RR-P3HT in Figure 1 as suggested by the reviewer and improved the schematic representation of dip coating and spin coating in the revised manuscript.

Reviewer 2, Minor 6: line 126 Super dehydrated hexane. please use dry hexane

Our response: We are thankful to the reviewer for this suggestion. We have changed the sentence in line 129 in the revised manuscript agreeing with the suggestion.

Reviewer 2, Comment 7:  line 142 "patterning source/drain electrodes using thermal evaporation" Please provide the thickness of Au source/drain

Our response: We are thankful to the reviewer for this suggestion. We have provided the thickness details in line 145 of the revised manuscript.

Reviewer 2, Comment 8: line 152 Please change "super dehydrated chloroform" by " dry chloroform"

Our response: Agreeing with the suggestion of the referee, we have made the suggested change in line 155 in the revised manuscript.

Reviewer 2, Comment 9: line 227 please change to improve the quality of the organic semiconductor film onto the dielectric surface which in turn particularly enhances its electrical performance"

Our response: We are thankful to the reviewer for this suggestion. We have reformulated the sentence as per the reviewer's suggestion in line 232 in the revised manuscript

Reviewer 2, Comment 10: Line 261 "chains of the P3HT" please keep the same abbreviation RR-P3HT.

Our response: We are thankful to the reviewer for this suggestion and sorry for our mistake. We have reformulated the sentence in line 265 in the revised manuscript.

Reviewer 2, Comment 11: Line 265 P3HT change to RR-P3HT

Our response: The suggestion has been incorporated as in line 270 in the revised manuscript.

Reviewer 2, Comment 12: Line 266 please delete this "(Fig. S1C. Please remove Figure S1B)."

Our response: We are thankful to the reviewer for this suggestion and sorry for our mistake. We have changed in line 269 as per the suggestion of the reviewer in the revised manuscript.

Reviewer 2, Comment 13: Line 276 the authors claimed that " the thickness of the film increases by decreasing the spinning speed and there was about 1.5 times increase" Can the authors provide a value of the effective thickness of the film for each speed?

Our response: Thank you very much for this concern and a similar concern has also been raised by reviewer 1. We have added the value of the effective thickness of the film for the decreasing spinning speed in lines 281-282 on page 7 in the revised manuscript.

Major comments

Reviewer 2, Comment 1: The authors attempt a quantitative analysis of crystallinity of RR-P3HT thin film from UV-vis absorption spectra of thin film fabricated by three different methods. The P3HT crystallinity can be obtained by comparing the two vibronic peaks at 555 nm and 610 nm using a Spano analysis. (Spano et al. Phys. Rev. Lett. 2007, 98.). Please add more detail to this analysis. In such a model the ratio of the two H-aggregates absorbance peaks is given by [(1 − 0.24W /Ep)/(1 + 0.73W /Ep)]2, where Ep = 0.18 eV is the energy of the symmetric C=C stretch, and W is the free exciton bandwidth within the crystalline phase. The ratio of the two vibronic peaks is related to the crystallinity phase in the film.

Our response: We thank the reviewer for pointing this out. We agree with the reviewer and newly added the information on exciton bandwidth in Table 1, Table 2, and Table 3 and also updated the necessary discussion in the revised manuscript. We have further evaluated the exciton bandwidth for all types of films and quantitatively discussed it in the thin film sections in lines 308 -311 of page 8, line 315 of page 8, lines 324-326 of page 9, lines 370-372 of page 10 and lines 453-455 of page 12 of the revised manuscript.

Reviewer 2, Comment 2: It is widely observed that coating P3HT using low speed in meniscus guided coating methods, such as deep-coating, yields to patterned films. (https://doi.org/10.1016/j.cis.2019.102080). The authors have used very low lifting speed, 20 to 200 μms-1 for the fabrication of P3HT film. My question is: Do the authors check the thin film morphology using optical or AFM microscopies?

Our response: Thank you very much for this valuable comment and suggestion. Authors agree with reviewer that the patterned films are formed while coating P3HT with meniscus guided methods at very low lifting speed. In fact, we also observed that by changing the surface treatment, the film patterning can be controlled. We have also observed patterned P3HT films using dip coating method as shown in figure B. Here, the films were fabricated at a lifting speed of 20 μms-1 on OTS treated substrates. On the other hand, in the present work, we optimized the thin films on the basis of their crystallinity and we found that the films having maximum crystallinity (on HMDS treated substrates) did not form patterns as shown in figure A. These optimized films were used for device fabrication. In this research, we did not discuss the patterned dip coated films since it is out of scope of this work. Please see the attachment for the figures.
